# Use of a Deep Learning Approach for the Sensitive Prediction of Hepatitis B Surface Antigen Levels in Inactive Carrier Patients

**DOI:** 10.3390/jcm11020387

**Published:** 2022-01-13

**Authors:** Hiroteru Kamimura, Hirofumi Nonaka, Masaya Mori, Taichi Kobayashi, Toru Setsu, Kenya Kamimura, Atsunori Tsuchiya, Shuji Terai

**Affiliations:** 1Division of Gastroenterology and Hepatology, Niigata University Graduate School of Medical and Dental Sciences, Niigata 951-8510, Japan; setsut@med.niigata-u.ac.jp (T.S.); kenya-k@med.niigata-u.ac.jp (K.K.); atsunori@med.niigata-u.ac.jp (A.T.); terais@med.niigata-u.ac.jp (S.T.); 2Department of Network Medicine for Digestive Diseases, Niigata University School of Medicine, Niigata 951-8510, Japan; 3Department of Information and Management System Engineering, Nagaoka University of Technology, Nagaoka 940-2188, Japan; nonaka@kjs.nagaokaut.ac.jp (H.N.); masaya.mori0905@gmail.com (M.M.); 4Division of Oral and Maxillofacial Radiology, Niigata University Graduate School of Medical and Dental Sciences, Niigata 951-8510, Japan; taichi@dent.niigata-u.ac.jp

**Keywords:** deep learning, machine learning, chronic hepatitis B, artificial intelligence

## Abstract

Deep learning is a subset of machine learning that can be employed to accurately predict biological transitions. Eliminating hepatitis B surface antigens (HBsAgs) is the final therapeutic endpoint for chronic hepatitis B. Reliable predictors of the disappearance or reduction in HBsAg levels have not been established. Accurate predictions are vital to successful treatment, and corresponding efforts are ongoing worldwide. Therefore, this study aimed to identify an optimal deep learning model to predict the changes in HBsAg levels in daily clinical practice for inactive carrier patients. We identified patients whose HBsAg levels were evaluated over 10 years. The results of routine liver biochemical function tests, including serum HBsAg levels for 1, 2, 5, and 10 years, and biometric information were obtained. Data of 90 patients were included for adaptive training. The predictive models were built based on algorithms set up by SONY Neural Network Console, and their accuracy was compared using statistical analysis. Multiple regression analysis revealed a mean absolute percentage error of 58%, and deep learning revealed a mean absolute percentage error of 15%; thus, deep learning is an accurate predictive discriminant tool. This study demonstrated the potential of deep learning algorithms to predict clinical outcomes.

## 1. Introduction

Risk prediction models are routinely used in healthcare practice to identify high-risk patients, guide treatment decisions, and effectively allocate healthcare resources according to patients’ needs. These risk prediction models are typically built using statistical regression techniques. Similar trends have also been reported for the clinical course of hepatitis B virus (HBV) [1,2].

Machine learning systems are used to identify objects in images, transcribe speech into text, match news items, match posts or products with user interests, and select relevant results in response to search requests. These applications increasingly make use of a class of techniques called deep learning (DL) [3]. DL is a branch of machine learning that employs algorithms to process data, imitate the thinking process, or develop abstractions and uses multiple layers of algorithms to process data. The application of DL models in the healthcare system has garnered great interest owing to the increasing complexity and volume of healthcare data [4].

Infection with the HBV leads to acute or chronic liver diseases. According to the World Health Organization report, 296 million individuals had chronic hepatitis B (CHB) in 2019; among them, 820,000 mortalities were recorded primarily owing to cirrhosis and hepatocellular carcinoma (HCC), which is a form of primary liver cancer. The clinical course of HBV infection is variable and includes acute (self-limiting) infections, fulminant hepatic failure, an inactive carrier state, and chronic hepatitis with potential progression to cirrhosis and HCC [5].

Chronic inflammation and subsequent hepatocarcinogenesis can be substantially affected by various factors associated with HBV. A study in Taiwan has revealed that hepatitis B surface antigen (HBsAg) or hepatitis B e antigen (HBeAg) is associated with hepatocarcinogenesis [6]. Hepatocarcinogenesis is reportedly high in those with high HBV DNA titer, and it is relatively higher when the amount of HBsAg is high in patients with low HBV DNA titer, indicating that the suppression of HBV replication and negative HBsAg are vital in deterring hepatocarcinogenesis. HBV replication has recently been well-controlled by nucleos(t)ide analog (NA) therapy. In such patients, the main antiviral therapy goals are HBV DNA suppression and alanine aminotransferase (ALT) level normalization [7]. However, in most patients, treatment does not cure HBV infection but only suppresses the replication of the virus. Therefore, most patients who start HBV treatment must continue it throughout their life. For patients with chronic hepatitis due to HBV infection, long-term suppressive NA therapy in HBeAg-positive patients has shown minimal effects in decreasing the HBsAg titer [8].

Notably, it is difficult to judge the efficacy of NA treatment based on the HBV DNA level, as this metric typically becomes less sensitive after antiviral treatment is initiated. Consequently, some reports have suggested that it is useful to alternatively monitor HBs antigen levels over time. In the case of HBeAg-positive chronic hepatitis B, the measurement of HBsAg levels after 24 weeks of treatment with Peg-IFNα-2a alone or in combination with lamivudine can be used to determine HBeAg seroconversion, HBV DNA levels, and HBsAg levels 24 weeks after the cessation of treatment [9]. Additionally, HBeAg seroconversion, HBV DNA content, and HBsAg reduction rate after 24 weeks of treatment can be predicted based on the measured HBsAg level at 24 weeks of treatment. In addition to predicting the efficacy of antiviral therapy, the need to measure HBsAg antigen over time in the natural course of HBV has been proposed.

A prospective study of HBV cases without prior antiviral therapy conducted in Taiwan revealed that the incidence of hepatocellular carcinoma is higher when the baseline HBV DNA levels are high (≥2000 IU/mL), and the incidence of hepatocellular carcinoma is lower in HBeAg-negative patients and those with low viral loads (<2000 IU/mL) [7]. By contrast, in HBeAg-negative and low viral load cases (<2000 IU/mL), the incidence of hepatocellular carcinoma was reported to be related to HBsAg levels [10]. Briefly, even if the HBV DNA level is <2000 IU/mL, the risk of carcinogenesis is high when the HBsAg level is ≥1000 IU/mL, and the risk is even higher in the group for whom an HBsAg level of ≥1000 IU/mL persists for more than three years [7].

The disappearance and reduction of HBsAg levels are primary treatment challenges worldwide [11]. In the past, many reports have highlighted the risks associated with high HBsAg levels; however, none have used DL to assess HBsAg levels. The reason for this is that HBsAg levels over a long time have not been uniformly obtained at the same institution [12,13,14].

DL can help predict the reduction of HBsAg levels with high sensitivity during outpatient care. Therefore, the aim of this study was to identify an optimal DL model to predict changes in HBsAg levels in daily clinical practice for inactive carrier patients.

## 2. Materials and Methods

### 2.1. Ethics Statement

The study protocol conformed to the guidelines of the Declaration of Helsinki and was approved by the Ethics Review Board of Niigata University (approval number 2020-0114).

This retrospective study enrolled patients who were not required to provide informed consent; only anonymous clinical data obtained from patients who consented to treatment were used in this study. Furthermore, we provided the option for patients to opt out of the study via a poster. The poster was approved by the Ethics Review Board of Niigata University.

### 2.2. Study Design, Participants, and Settings

This retrospective, observational, single-center study was conducted at the Niigata University Hospital and only included patients who were HBsAg-positive.

Patients who had HBsAg examinations between April 2003 and December 2019 were screened. The inclusion criteria were as follows: a confirmed record of HBsAg quantitative levels and ALT, aspartate transaminase (AST), and HBV DNA levels measured in the first, second, fifth, and tenth years; not receiving NA therapy in the observation period; and no development of hepatocellular carcinoma.

### 2.3. Sample Analysis

Serum HBsAg titers were measured using Lumipulse Ii HBsAg (Fujirebio Inc., Tokyo, Japan) from 2003 to 2007 and Architect HBsAg QT (Abbott Laboratories, Rungis, France) from 2008 onwards. The correlation of HBsAg assay kits has been reported previously and does not introduce errors in the generation of teacher data [15]. HBV DNA levels were quantified using real-time PCR with the TaqMan PCR system (Roche Diagnostics, Tokyo, Japan).

### 2.4. Statistical Analysis

Continuous variables are expressed as the median and interquartile range, and differences were assessed using the chi-square test, Fisher’s exact test, or Mann–Whitney U test, as appropriate. The correlation among continuous variables was assessed using linear regression analysis. All analyses were performed using IBM SPSS Statistics for Windows, version 22.0 (IBM Corp., Armonk, NY, USA). Statistical significance was defined as *p*-value < 0.05.

### 2.5. Deep Neural Network (DNN)

To construct and modify the machine learning model, a Windows PC, with Intel Core i5 2 GHz carrying 16 GB RAM, and NNC version 1.10 (Sony Corp., Tokyo, Japan) were used as a DL-integrated development environment.

### 2.6. Dataset Creation

After creating the dataset, the “input folder” and “output folder” were first created, and data were categorized in folders and stored in the “input folder.” Subsequently, data regarding the age, weight, height, sex, ALT level, AST level, HBV DNA level, and HBsAg level after 1, 2, 5, and 10 years for each patient were set on the DATASET creation screen of the NNC. The ratio of training data to test data was specified and executed to a comma-separated values (CSV) file of the dataset that was created.

The large amount of data contained in a dataset is generally divided into two subsets. One subset is used for training, and the other is used to evaluate the performance of the DL model after training. Approximately 20% of the evaluation data was allocated for verification. At that time, we also ensured that the combination of training data and evaluation data was normally distributed. In addition, as explained in the inclusion criteria section, only cases with no missing values for each item were included.

### 2.7. Network Configuration

The DL program allows for easy editing of the configuration using drag and drop options; it also allows users to design neural networks using multiple layers. New ideas can be implemented in a matter of seconds, allowing better performance and automatic identification of lighter neural networks. For this analysis, we built a three-layer network, comprising an input layer, a hidden layer, and an output layer. The network was constructed by dragging and dropping components from the component list on the left side of the network graph (Figure 1).

In the project screen of the NNC, a one-layer neural network (Input, Affine, Rectified Linear Unit (ReLU), and Affine2) was created by adding functions to the network graph, starting with Input in components. Thereafter, we checked if the CSV file of the dataset could be read.

Affine was used as an all-attachment layer that combines all input values to all output layers as specified by the out-shape property. ReLU was used as a function in which the output value was always zero when the input value of the function was ≤0, and the output value was the same as the input value when the input value was >0. After the second layer of Affine, the loss function HuberLoss_2 was set at the end. Huber Loss was employed as a loss function to help detect small errors using squared error and large errors using absolute error; thus, small errors were weighted, and large irregular errors were less weighted.

### 2.8. Evaluation

By clicking on the learning curve button to commence learning, the progress was illustrated in the Learning Curve on the graph monitor (the learning result screen). The training ended with the set number of epochs (Figure 2). The evaluation result could be checked in the Evaluation tab, and the evaluation operation commenced when the Confusion Matrix tab was clicked and “estimated y” was displayed (Figure 3).

One of the most common metrics used to measure the forecast accuracy of a model uses the mean absolute percentage error (MAPE) as follows:$$\mathrm{MAPE}\;=\;\left(1/n\right)\;\times \;\Sigma \;\left(\left|\mathrm{actual}\;-\;\mathrm{forecast}\right|/\left|\mathrm{actual}\right|\right)\;\times \;100,$$
where n is the sample size; actual, the actual data value; and forecast, forecasted data value. MAPE is commonly used because its results are easy to interpret and explain. MAPE was designed for use in both statistical analyses and NNCs.

## 3. Results

### 3.1. Patient Characteristics

There were 1820 HBsAg-positive patients identified in the initial screening (Figure 4). A total of 1583 patients without HBsAg levels for years 1, 2, 5, and 10 were excluded. A further 147 patients who underwent nucleic acid analog therapy during the investigation period were excluded from the analysis to prevent differentiation between inactive carrier and NA treatment patients as Boglione et al. reported that the calculated expected time to HBsAg loss is shorter for tenofovir than for telbivudine [16].

The training data comprised data from 72 patients, and the evaluation data included 18 patients with no missing values for HBsAg, ALT, and AST levels after 1, 2, 5, and 10 years and HBV DNA titer after 1 year. The characteristics of the patient training data are presented in Table 1. There were no significant differences between the training data and evaluation data.

### 3.2. Evaluation of Statistical Analysis

#### Multivariate Logistic Regression Model

In the initial analysis, 17 items were examined, and 17 continuous variables were used. In the multivariable analysis model, nine variables, including HBeAg- rate, HBs levels after 5 years, and HBs levels after 2 years, were included. The multiple correlation coefficient was 0.989, and the coefficient of determination (R^2^) was 0.97. Furthermore, the logistic regression model generated the coefficients of a formula to predict a logit transformation of the probability of the presence of the characteristic of interest as follows: HBsAg at 10 years = −1167 + −155 (HBe positive) − 35.6 (HBV DNA first year) + 0.017 (HBsAg first year) + 12.2 (ALT first year) + −0.736 (HBsAg second year) + 1.215 (HBsAg fifth year) + −9.955 (ALT fifth year) (Table 2).

Multiple regression analysis was performed to assess serum HBsAg levels. We compared these values to the predicted value of serum HBsAg levels after 10 years, determined using DL. Multiple regression analysis revealed a MAPE of 58%.

### 3.3. Evaluation of DL

#### Implementation and Evaluation of Learning

For the 18 patients included in the validation dataset for which antigen levels after 10 years were known, the CSV file of the learning results was referred to as the evaluation item after executing supervised data learning, and the predicted values were calculated a′ y′.

Using the DNN, the new validation for 18 patients showed an MAPE of 15%. When verified using the mean absolute error rate, the model created using DNN was accurate, partly because it had a positive value.

The accuracy of the predicted value from the DL and multivariate logistic regression models are shown Figure 5. The correct median of the evaluation data and the predicted value from DL and multivariate logistic regression model are calculated as follows.
MAPE = (1/n) × Σ(|correct median of Evaluation data − Predicted value from DL or Multivariate logistic regression model|/|correct median of Evaluation data|) × 100,
where n is the sample size.

## 4. Discussion

In recent years, the evolution of artificial intelligence (AI) has been remarkable, and this is due to the emergence of DL—a type of machine learning, which is a technology that allows computers to learn in a similar manner to the human brain. To extract features, neural networks that mimic the mechanism of human neurons are used, and the hierarchy of calculations for deriving results is deeper than that used for previous AI models [17].

There are two reasons why DL has become common: the first is the spread of the internet and the miniaturization of various sensors, which have made it possible to obtain large amounts of data that can be used for AI learning; the second is the emergence of graphics processing units (semiconductors for image processing), which have a considerably higher capacity for parallel operations than central processing units. The neural networks that form the basis of DL had already been devised, but they showed insufficient processing performance. With the aid of a large amount of data and high parallel computing power, the practical application of DL progressed rapidly [18].

Here, we applied a DNN technique to evaluate future decreased HBsAg levels and then compared its performance with a traditional statistical model. We observed that the efficacy of the DNN method was comparable to that of the traditional statistical model. DNN techniques have been used to predict factors, including physicochemical properties.

HBsAg loss is considered the ideal therapeutic goal in HBV-infected patients; however, it is rarely achieved via treatment with the currently available antiviral agents [19]. HBsAg seroclearance is widely considered to be an important indicator of CHB prognosis [20]. The spontaneous seroclearance of HBV DNA and HBsAg is an important predictor of reduced HCC risk [21]. Thi Vo et al. analyzed data from 1840 patients who presented with HCC; among them, 75.5% (1390) had high HBsAg levels, at over 1000 (IU/mL), and 24.5% (450) had low HBsAg levels, at under 1000 (IU/mL). The participants had a 2.46-fold rise in the risk of HCC development compared with those with lower HBsAg levels [22]. An understanding of HBsAg titer changes throughout HBV infection may provide some potentially valuable insights into hepatitis B pathogenesis and viral life cycle [23]. Although multifactor analysis has been previously performed, reliable predictors of the natural history of disappearance or decreased HBs levels have not been established. An additional point to consider is that the HBsAg level is also affected by factors such as age, HBV DNA level, and HBV genotype [24].

HBeAg and HBsAg titers have been proposed as biomarkers for infected liver cell mass or HBV covalently closed circular DNA (cccDNA), the hepatocyte nuclear reservoir that is responsible for viral persistence [25]. This concept supports their use as biomarkers. If the hepatocyte is considered in isolation, HBsAg, HBeAg, and serum HBV DNA levels would be expected to directly correlate with each other and with liver cccDNA levels, as all are translated from separate transcripts (Pre-S1, Pre-S2/S, precore/core, and pre-genomic mRNA, respectively) directly derived from cccDNA. However, the published data that describe these relationships are limited and conflicting [26,27]. ALT levels were included in the logistic regression model, and the result matches that reported previously [28].

In patients treated with NA, it also declines very slowly, even though serum HBV DNA levels decrease significantly. Low serum HBsAg may predict either spontaneous or treatment-induced HBsAg seroclearance and potentially selects HBeAg-negative patients who can safely stop NA. High serum HBsAg is associated with a high risk of hepatocellular carcinoma in an untreated population and predicts treatment failure in patients receiving pegylated interferon. Therefore, the potential roles of HBsAg quantification are applicable to selected populations only. There is also a need for novel markers to study the effect of emerging antiviral therapies targeting various parts of the HBV cycle to reflect their distinct mechanistic effects. Several agents measuring HBsAg levels have shown a rapid and significant decline. Ongoing studies are required to demonstrate the sustainability of HBsAg suppression by these novel agents [29].

In the multiple regression equation generated from the statistics, if a factor is not significant, it is deleted as an item; however, the DL reflects it even if it is not significant, which may indicate why the MAPE showed improved accuracy in the DL model compared to the logistic regression model. In the future, when additional cases are added, or when new markers are discovered, the error can be continually improved by adding new items to the DL, without needing to perform complicated multiple regression equations.

The use of DNN algorithms to predict disease status or outcomes in clinical datasets is consistently gaining more attention in medicine and healthcare, as reported by a study that reviewed relevant topics [30]. In this retrospective cohort study, we compared HBsAg levels predicted using DL and those predicted using statistical analysis.

In terms of the prediction of viral spread using AI, there are several leading reports involving COVID-19. The prediction of coronavirus infections using DNN has been reported [31], and DNN is likely to be the best model for predicting trends in viral diseases that continually show diversity and are influenced by multiple factors. There are several advantages of using AI to fight against pathogens such as viruses and bacteria that cause a complex reaction in the host [32]. DNN can accelerate the discovery of valuable vaccines or drugs to prevent pandemics and facilitate the diagnosis of diseases [33,34].

There are some limitations to this study, including the number of participants required for the training data. Entecavir, tenofovir disoproxil fumarate, and tenofovir alafenamide were approved for the treatment of CHB by the Food and Drug Administration in 2005, 2008, and 2016, respectively [35]. However, we did not include patients who were receiving nucleic acid analog therapy at the time of analysis to remove uncertainty in the training data.

DNN requires a specific area of expertise; however, this study demonstrates that clinicians can make accurate predictions using simple DL tools, rather than producing complex statistical formulas, using data managed by their institutions.

## Figures and Tables

**Figure 1 jcm-11-00387-f001:**
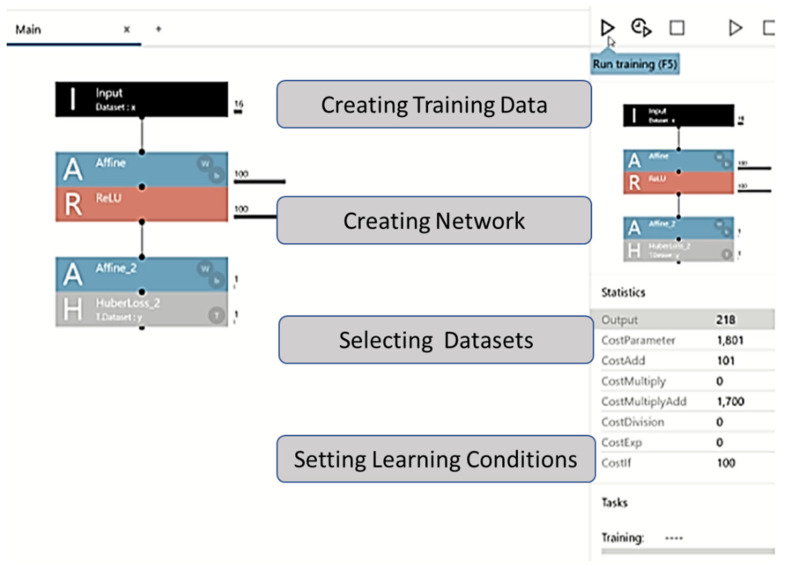
Graphical user interface of the machine learning framework. The figure explains the experimental process with the Neural Network Console. Learning progress and learning results screens are shown. After setting the parameters, one can set the training conditions, the number of epochs (Max Epoch), and batch size (Batch Size) in the Global Config on the CONFIG tab.

**Figure 2 jcm-11-00387-f002:**
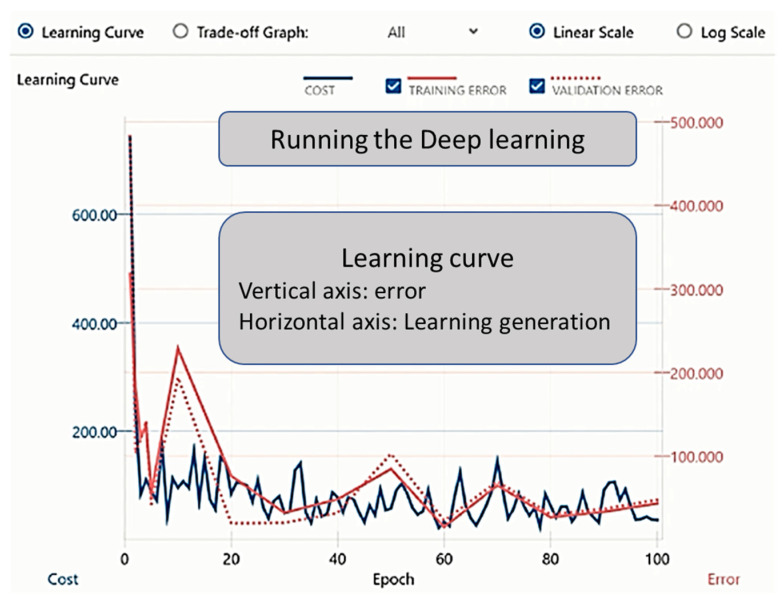
Graphical user interface of the DL process. The figure explains the experimental process with the Neural Network Console. The “Learning Progress” and “Learning Results” screens are shown. Clicking on the learning curve button commences deep learning, and the progress will be shown in the Learning Curve on the graph monitor (the learning result screen). Training ends with the set number of epochs.

**Figure 3 jcm-11-00387-f003:**
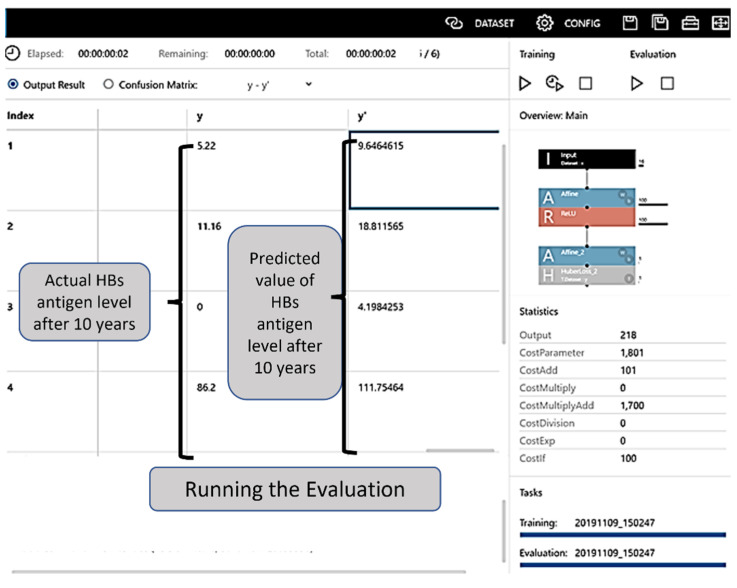
Evaluation table. The evaluation result can be checked in the Evaluation tab, and the evaluation is initiated when the Confusion Matrix tab is clicked and “estimated y” is displayed. HBs: hepatitis B surface.

**Figure 4 jcm-11-00387-f004:**
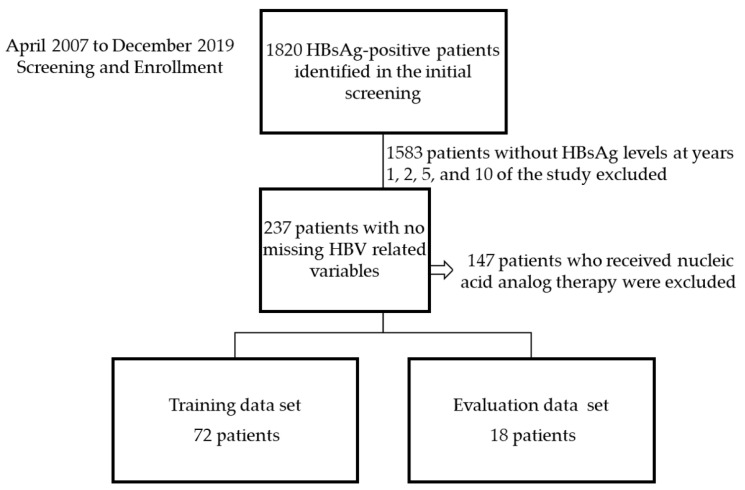
Flow chart of patient selection. HBV, hepatitis B virus; HBsAg, hepatitis B surface antigen.

**Figure 5 jcm-11-00387-f005:**
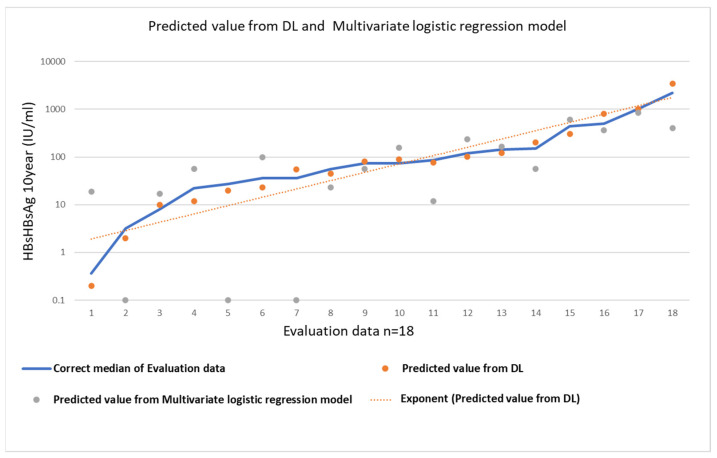
Evaluation table. The correct median of the evaluation data and the predicted value from DL and multivariate logistic regression model.

**Table 1 jcm-11-00387-t001:** Baseline characteristics of participants enrolled in the study of training data and evaluation data.

Continuous Survey over 10 Years (*n* = 90); Median (Range or %)			
Variable	Training Data (*n* = 72)	Evaluation Data (*n* = 18)	*p*-Value
Sex, male (%)	69 (63)	13 (65)	n.s.
Age (y)	69 (14–78)	67 (20–81)	n.s.
HT (cm)	162 (123–189)	165 (144–177)	n.s.
Body weight (kg)	59 (33–98)	61 (39–88)	n.s.
HBeAg negative rate (%) *	67/72 (92)	16/18 (93)	n.s.
HBV DNA first year (Log(IU/mL))	3.2 (1.9–5.2)	3.1 (2.2–5.4)	n.s.
HBsAg first year (IU/mL)	455 (45–7202)	421 (42–5915)	n.s.
ALT first year (IU/L)	33 (15–42)	31 (21–49)	n.s.
AST first year (IU/L)	28 (7–51)	22 (9–45)	n.s.
HBV DNA second year (Log(IU/mL))	2.7 (1.5–5.0)	2.9 (1.9–5.1)	n.s.
HBsAg second year (IU/mL)	233 (7.2–4949)	201 (9.3–4233)	n.s.
ALT second year (IU/L)	29 (11–39)	31 (13–45)	n.s.
AST second year (IU/L)	26 (19–48)	23 (11–45)	n.s.
HBV DNA fifth year (Log(IU/mL))	2.5 (1.4–4.3)	2.8 (2.1–4.9)	n.s.
HBsAg fifth year (IU/mL)	112 (0.7–4025)	101 (0.9–3336)	n.s.
ALT fifth year (IU/L)	21 (9–46)	26 (9–44)	n.s.
AST fifth year (IU/L)	26 (19–45)	26 (11–49)	n.s.
HBV DNA tenth year (Log(IU/mL))	2.5 (N.D–4.3)	2.8 (N.D–4.9)	n.s.
HBsAg tenth year (IU/mL)	98 (0.1–4551)	78 (0.4–2221)	n.s.
ALT tenth year (IU/L)	19 (7–39)	21 (8–56)	n.s.
AST tenth year (IU/L)	22 (12–35)	21 (11–46)	n.s.

HBV, hepatitis B virus; HBsAg, hepatitis B surface antigen; HBeAg, hepatitis B envelope antigen; AST, aspartate aminotransferase; ALT, alanine aminotransferase; n.s., not significant; N.D, not detected. * HBeAg was not determined for two cases.

**Table 2 jcm-11-00387-t002:** Variables that were included in the logistic regression model.

Factors for Predicting Serum HBsAg 10 Years after HBs Titer According to Univariable and Multivariable Analyses								
Variable	Univariable Analysis				Multivariable Analysis			
	OR	95% CI		*p*	OR	95% CI		*p*
Constant					−1167			
Sex	28.378	−58.158	114.914	0.514				
HT, cm	5.809	0.963	10.655	0.020	6.942	4.068	9.816	0.068
Body weight, kg	1.183	−1.659	4.024	0.408				
Age, y	−1.214	−4.302	1.873	0.434				
HBeAg negative rate, % *	−142.964	−244.111	−41.816	0.006	−154.811	−255.179	−54.442	0.003
HBV DNA first year, Log (IU/mL)	−40.079	−70.535	−9.623	0.011	−35.566	−64.631	−6.502	0.017
HBsAg first year, IU/mL	0.278	0.209	0.347	0.000	0.117	0.060	0.175	0.000
ALT first year, IU/L	12.037	6.186	17.889	0.000	12.273	7.286	17.261	0.000
AST first year, IU/L	−5.747	−24.151	12.657	0.536				
HBV DNA second year, Log (IU/mL)	−20.074	−51.535	−7.623	0.068				
HBsAg second year, IU/mL	−0.756	−0.880	−0.632	0.000	−0.736	−0.843	−0.628	0.000
ALT second year, IU/L	7.512	−18.602	33.627	0.568				
AST second year, IU/L	22.523	−1.205	46.250	0.062	0.675	−1.788	3.139	0.586
HBV DNA fifth year, Log (IU/mL)	−18.156	−50.424	−8.362	0.075				
HBsAg fifth year, IU/mL	1.218	1.126	1.310	0.000	1.215	1.135	1.295	0.000
ALT fifth year, IU/L	−12.791	−19.521	−6.060	0.000	−9.955	−15.016	−4.894	0.000
AST fifth year, IU/L	−1.363	−7.135	4.409	0.638				

* 2 cases were not performed HBeAg.

## Data Availability

The data underlying this article are available in this article.

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
