# Peer review of "Use of a Deep Learning Approach for the Sensitive Prediction of Hepatitis B Surface Antigen Levels in Inactive Carrier Patients"

_jcm, 2022, doi:10.3390/jcm11020387_

Round 1

Reviewer 1 Report

The paper deal with the use of a Deep Learning (DL) approach for the sensitive prediction  of hepatitis B surface antigen levels in inactive carrier patients. DL is a branch of machine learning that employs algorithms to process data.  The application of DL  in the healthcare system has garnered great interest owing to the increasing complexity and volume of healthcare data. In this context, the authors tried to identify an optimal DL model to predict changes in HBsAg levels in daily clinical practice for inactive carrier patients. The topic is certainly important, however, in the reviewer’s opinion the paper as it now stands is not ready to be published, as there are important aspects to be re-analysed in the work.

Materials and Methods:

  • Section 2.2 Study Design, Participants, and Settings line 86: please indicated the total number of patients excluded from the analyses
  • Section 2.2 Study Design, Participants, and Settings lines 86-88: please indicate it is not indicated how many patients have started antiviral treatment during the follow-up. If there are any, please provide relevant data.
  • Section 2.6 Creating Datasets: the construction of the " training data" is not clear, additional comments need to be provided.

Results:

  • Section 3.1. Patient Characteristics: lines 161-162 and table 1: the selection of the patients needs further clarification. Also the clinical characteristics of patients during the follow-up are missing and need to be included.  

  • Previous studies (Mak LY et al Hepatol Int. 2020 Jan;14(1):35-46) indicate that serum levels of HBsAg in HBeAg-negatives patients with chronic hepatitis B are not associated (barely linearly correlated) with replication markers of HBV. On the contrary in the present paper those quantities are correlated in some sense. It would be meritorious if the authors would provide comments on these aspects.

As concluding remark despite the DL approach might provide interesting and novel results, in the reviewer opinion it is imperative to validate and to verify those results in a clinical context. The authors are invited to consider the clinical outcome of the patients also in relation with previous literature.

Author Response

29 Dec 2021

Dr. Emmanuel Andrès

Editor-in-Chief

Journal of Clinical Medicine

Dear Editor:

Please find attached a revised version of our manuscript for publication in the Journal of Clinical Medicine, titled “Use  of a Deep Learning Approach for the Sensitive Predic-tion of Hepatitis B Surface Antigen Levels in Inactive Carrier Patients” The paper was coauthored by Hirofumi Nonaka, Masaya Mori, Taichi Kobayashi, Toru Setsu, Kenya Kamimura, Atsunori Tsuchiya, and Shuji Terai. The manuscript ID is jcm-1528128.

 Thank you very much for your e-mail and review of the manuscript. Your comments and those of the reviewers were highly insightful and enabled us to improve the quality of our manuscript. We added and revised some articles in the discussion section of the manuscript. We marked up using the “Track Changes” function. We responded to all the comments from the reviewers as indicated below. We reproofed the article to Editage editing survice.

We hope that the revisions in the manuscript and our accompanying responses will be sufficient to make our manuscript suitable for publication.

We look forward to hearing from you regarding our submission. We would be glad to respond to any further questions and comments that you may have.

We appreciate the comments of Reviewer, which we have taken account in the revised manuscript. Below are our responses to the concerns of this reviewer.

Author's Reply to the Review Report (Reviewer 1)

The paper deal with the use of a Deep Learning (DL) approach for the sensitive prediction  of hepatitis B surface antigen levels in inactive carrier patients. DL is a branch of machine learning that employs algorithms to process data.  The application of DL  in the healthcare system has garnered great interest owing to the increasing complexity and volume of healthcare data. In this context, the authors tried to identify an optimal DL model to predict changes in HBsAg levels in daily clinical practice for inactive carrier patients. The topic is certainly important, however, in the reviewer’s opinion the paper as it now stands is not ready to be published, as there are important aspects to be re-analysed in the work.

We appreciate the comments of Reviewer, we respond to the concerns of this reviewer.

  • Materials and Methods:Section 2.2 Study Design, Participants, and Settings line 86: please indicated the total number of patients excluded from the analyses

Response: We added the Study Design, Participants, and Settings Section 2.2 and described Rescult section 3.1.

  • Section 2.6 Creating Datasets: the construction of the " training data" is not clear, additional comments need to be provided.

Response: We added the description of the 2.6 Dataset Creation in line 141-146 as reviewer suggested.

  • Results: Section 3.1. Patient Characteristics: lines 161-162 and table 1: the selection of the patients needs further clarification. Also the clinical characteristics of patients during the follow-up are missing and need to be included.  

Response: We added the Figure 4 for description of flow chart of patient selection as the reviewer suggested. 

5)Previous studies (Mak LY et al Hepatol Int. 2020 Jan;14(1):35-46) indicate that serum levels of HBsAg in HBeAg-negatives patients with chronic hepatitis B are not associated (barely linearly correlated) with replication markers of HBV. On the contrary in the present paper those quantities are correlated in some sense. It would be meritorious if the authors would provide comments on these aspects.

Response: We added the references as the reviewer suggested Mak LY et al Hepatol Int. 2020 Jan;14(1):35-46 in reference 28 and there are opposite  study Alghamdi, A et al. Saudi J Gastroenterol. 2013, 19, 252–257 which suggested an additional point to consider is that the HBsAg level is also affected by factors such as age, HBV DNA level, and HBV genotype as reference 24. We described compaed to our data in line 288-308.

Thank you for your consideration. I look forward to hearing from you.

Sincerely,

Hiroteru Kamimura, M.D., Ph.D.

Reviewer 2 Report

In the present manuscript, Kamimura and colleagues described a deep learning approach for the prediction of HBsAg levels variation in patients with HBeAg-negative chronic infection (i.e. inactive carriers). The topic is interesting and the idea is novel; however, the manuscript needs to be improved in order to be more understandable to readers not confident to such approaches. 

Below, the specific comments:

1) Introduction. lines 36. Please add some references. For instance, the authors may consider a previously published paper on JCM (PMID: 34362093).

2) Materials and Methods. 2.3 Sample analysis. HBsAg was measured on samples collected between 2003 and 2007 by Lumipulse and from 2008 by Architect. Were these methods consistent? Do you have any data on concordance between the methods? If not available, please add a reference on this aspect.

3) The authors detaily described the protocol of the deep learining approach; however, it could be useful for the readers to describe more the method rather than the protocol. Indeed, this article is of partcilular interest to clinicians that are usually not confident with this approach.

4) Results. This section must be improved; please describe what is MAPE and improve the comparison between multiple regression analysis and DL. it is really difficult to understand and the reader can not perceive the added value of DL compared to standard statistical approach.

Author Response

29 Dec 2021

Dr. Emmanuel Andrès

Editor-in-Chief

Journal of Clinical Medicine

Dear Editor:

Please find attached a revised version of our manuscript for publication in the Journal of Clinical Medicine, titled “Use  of a Deep Learning Approach for the Sensitive Predic-tion of Hepatitis B Surface Antigen Levels in Inactive Carrier Patients” The paper was coauthored by Hirofumi Nonaka, Masaya Mori, Taichi Kobayashi, Toru Setsu, Kenya Kamimura, Atsunori Tsuchiya, and Shuji Terai. The manuscript ID is jcm-1528128.

 Thank you very much for your e-mail and review of the manuscript. Your comments and those of the reviewers were highly insightful and enabled us to improve the quality of our manuscript. We added and revised some articles in the discussion section of the manuscript. We marked up using the “Track Changes” function. We responded to all the comments from the reviewers as indicated below. We reproofed the article to Editage editing survice.

We hope that the revisions in the manuscript and our accompanying responses will be sufficient to make our manuscript suitable for publication.

We look forward to hearing from you regarding our submission. We would be glad to respond to any further questions and comments that you may have.

We appreciate the comments of Reviewer, which we have taken account in the revised manuscript. Below are our responses to the concerns of this reviewer.

Author's Reply to the Review Report (Reviewer 2)

In the present manuscript, Kamimura and colleagues described a deep learning approach for the prediction of HBsAg levels variation in patients with HBeAg-negative chronic infection (i.e. inactive carriers). The topic is interesting and the idea is novel; however, the manuscript needs to be improved in order to be more understandable to readers not confident to such approaches. 

We appreciate the comments of Reviewer, we respond to the concerns of this reviewer.

Below, the specific comments:

 1)Introduction lines 36. Please add some references. For instance, the authors may consider a previously published paper on JCM (PMID: 34362093).

Response: We added the references as the reviewer suggested in introduction as reference 1.

2) Materials and Methods. 2.3 Sample analysis. HBsAg was measured on samples collected between 2003 and 2007 by Lumipulse and from 2008 by Architect. Were these methods consistent? Do you have any data on concordance between the methods? If not available, please add a reference on this aspect.

Response: We added the Study Design, Participants, and Settings Section 2.2 and described Rescult section 3.1 and added the Figure 4 for description of flow chart of patient selection as the reviewer suggested.  We added the references as the reviewer suggested as “The correlation of HBsAg assay kits has been reported previously and does not intro-duce errors in the generation of teacher data [15.Murayama, A.; Momose, H.; Yamada, N.; Hoshi, Y.; Muramatsu, M.; Wakita, T.; Ishimaru, K.; Hamaguchi, I.; Kato, T. Evaluation of in vitro screening and diagnostic kits for hepatitis B virus infection. J Clin Virol. 2019, 117, 37-42. https://doi.org/10.1016/j.jcv.2019.05.011.]”

3) The authors detaily described the protocol of the deep learining approach; however, it could be useful for the readers to describe more the method rather than the protocol. Indeed, this article is of partcilular interest to clinicians that are usually not confident with this approach.

Response: We added the protocol in 2.7 Network Configuration as “The DL program allows for easy editing of the configuration using drag and drop options; it also allows users to design neural networks using multiple layers. New ideas can be implemented in a matter of seconds, allowing better performance and auto-matic identification of lighter neural networks. For this analysis, we built a three-layer network, comprising an input layer, hidden layer, and output layer. The network was constructed by dragging and dropping components from the component list on the left side of the network graph (Figure 1).eferences as the reviewer suggested.”

 Results. This section must be improved; please describe what is MAPE and improve the comparison between multiple regression analysis and DL. it is really difficult to understand and the reader can not perceive the added value of DL compared to standard statistical approach.

Response: We revised the list the specific items, and added 3.3.1 Implementation and evaluation of learning as “The accuracy of the predicted value from the DL and multivariate logistic regression models are shown Figure 5. The correct median of the evaluation data and the predicted value from DL and multivariate logistic regression model are calculated as follows.

MAPE=(1/n)×Σ (|correct median of Evaluation data–Predicted value from DL or Multivariate logistic regression model|/|correct median of Evaluation data|)×100,

where n is the sample size.”

Thank you for your consideration. I look forward to hearing from you.

Sincerely,

Hiroteru Kamimura, M.D., Ph.D.

Round 2

Reviewer 2 Report

The authors appropriately revised the manuscript according to the comments raised.